# Retrospective Analysis of Sterile Corneal Infiltrates in Patients with Keratoconus after Cross-Linking Procedure

**DOI:** 10.3390/jcm11030585

**Published:** 2022-01-25

**Authors:** Magdalena Krok, Ewa Wróblewska-Czajka, Joanna Kokot, Anna Micińska, Edward Wylęgała, Dariusz Dobrowolski

**Affiliations:** 1Chair and Clinical Department of Ophthalmology, Faculty of Medical Sciences in Zabrze, Medical University of Silesia, Panewnicka 65 Street, 40-760 Katowice, Poland; ewaw8@wp.pl (E.W.-C.); lek.joanna.kokot@gmail.com (J.K.); anna.micinska93@gmail.com (A.M.); wylegala@gmail.com (E.W.); dardobmd@wp.pl (D.D.); 2Ophthalmology of Department, District Railway Hospital, Panewnicka 65 Street, 40-760 Katowice, Poland

**Keywords:** sterile corneal infiltrate, cross-linking, keratoconus

## Abstract

Background: This paper’s objective is to analyze patients with keratoconus who developed sterile infiltrate after corneal collagen cross-linking (CXL), and to evaluate possible risk factors for their occurrence. Methods: 543 medical histories of patients after cross-linking (Epi-off, Epi-on) procedure performed according to the Dresden protocol were analyzed retrospectively. Results: Sterile corneal infiltrates occurred in four men (0.7%) in the age range (16–28) years, the average age being 20.3. The average time from procedure to onset of symptoms was 3.5 days (2–5 days). Inflammatory infiltration resolved in all patients, leaving scars on corneal stroma in two patients. Corneal healing time ranged from 4–12 weeks. In vivo confocal microscopy (IVCM), round inflammatory cells, and Langerhans cells in the epithelium and Bowman’s layer were observed at the site of infiltration. The Optical coherence tomography (OCT) shows hyperreflective lesions of various sizes which decreased over time. The corneal topographic parameters and Best-corrected visual acuity (BCVA) improved after the CXL procedure in all of the described cases. Conclusions: Most likely, damage to the epithelium and the phototoxic effect of the procedure is of significant importance in the formation of sterile corneal infiltrates. Appropriate classification and selection of CXL procedures in combination with protective measures in people at risk may have an overwhelming impact on the incidence of this complication.

## 1. Introduction

Keratoconus is the most common corneal ectasia. In the course of this disease, the central and paracentral parts of the cornea become thinned and, as a consequence, develop irregular bulging. As a result, the curvature of the cornea takes on a conical shape, causing irregular astigmatism, which reduces visual acuity and causes ineffectiveness of the prescribed eyeglass correction [1]. A method of treatment with proven effectiveness is the cross-linking of collagen fibers within the cornea (Corneal cross-linking, CXL), first described by Wollensak et al. [2]. The aim of this conventional procedure is to increase the stability of the corneal collagen fibers using riboflavin and UV radiation. The most common CXL procedure is based on the so-called Dresden protocol [2,3].

CXL is a very popular procedure used to inhibit the progression of keratoconus. This procedure has a high safety profile, however, interference with corneal tissue by removing the epithelium followed by exposure to the UV light source may lead to complications involving the surface of the eye and the corneal stroma. One of the rare complications is sterile keratitis. This disorder also occurs after refractive procedures, where interference with normal tissue results in an inadequate inflammatory reaction with the possibility of unfavorable tissue remodeling and loss of its optical properties [4,5,6]. 

The etiology of sterile corneal infiltrates is still unclear. Few papers describing this complication have been published over the years. It is suspected that the following factors contribute to the formation of sterile infiltrates: increased response to staphylococcal antigens that accumulate in tears at the eye shield contact, causing an increased immune response [7], epithelial damage, vernal keratoconjunctivitis [8], phototoxicity of the procedure [9], thinner and steeper cornea before the procedure [10], and postoperative use of NSAIDs [11,12]. Recent studies indicate increased incidence in people with Down’s syndrome [8]. Clarification of what may be the cause of this complication will reduce the number of patients with this condition and allow for faster diagnosis and implementation of the appropriate treatment, which is different in infectious keratitis.

The aim of this paper is the retrospective analysis of four patients in whom sterile infiltrate occurred after CXL surgery at the Department of Ophthalmology, Medical University of Silesia in Katowice in 2011–2020. In the text, the authors describe the course of the disease, diagnostics and medical procedures in these patients. They analyze the causes and compare them with previous reports.

## 2. Materials and Methods

### 2.1. Study Population

The study was conducted in accordance with the guidelines of the Declaration of Helsinki, and approved by the Bioethical Committee of the Medical University of Silesia, Katowice, Poland (PCN/0022/KB1/21/21). Medical histories of 543 patients who underwent corneal cross-linking (CXL) procedures due to progressive keratoconus in 2011–2020 were analyzed retrospectively. The analyzed group included 312 men and 231 women aged 22.7 years on average (14–47 years). Average follow-up period was 6.7 years (1–9 years). Criteria for progression of the disease included changes in the corneal topographic parameters, such as: increase in astigmatism or corneal curvature (K1, K2), maximum corneal curvature (Kmax) by >1D within 12 months from the last follow-up visit, increase in corneal elevation by >15 µm and deterioration of visual acuity by one or more Snellen lines in corrected visual acuity. The mjority of the patients used contact lenses before the procedure; it was recommended to discontinue their use 14 days before the planned procedure.

### 2.2. Examinations

All patients underwent the procedure, and depending on cornea thickness, the following was performed: pachymetry >400 µm—Epi-off procedure (3 mW/cm^2^, 30 min), and with corneal thickness <400 µm—Epi-on (3 mW/cm^2^, 30 min). Both were in accordance with the Dresden protocol. After the procedure, 1 drop of levofloxacin (5 mg/mL) and dexamethasone (1 mg/mL) were administered and a soft eye shield (Air Optix Aqua; Ciba Vision, Alcon, Fort Worth, TX, USA, 14.2 diameter, 8.6 base curve) was applied in a sterile way for 7 days until the first check-up. According to the clinic’s procedure, the first check-up was scheduled after 7 days, unless adverse symptoms occurred, in which case the patient was advised to come to the center as soon as possible. Each patient received detailed instructions on how to proceed with eye care after the procedure, along with hygienic recommendations. All patients used levofloxacin and dexamethasone drops and preservative-free artificial tears (based on sodium hyaluronate) five times a day until the first check-up after 7 days. When a suspicion of keratitis occurred, the treatment was appropriately modified, and a swab was taken for microbiological examination. 

## 3. Results

Four cases of sterile corneal infiltrate were found in the analyzed group. They were men in the age range (16–28) years, the average age being 20.3. The overall incidence rate in the centre was 0.7%. The first check-up was scheduled 7 days after the surgery. Only one patient came in a day earlier than the designated appointment; the rest of the patients attended on the appointed date reporting unrelenting redness, pain in the eye that had undergone the procedure, and progressive deterioration of vision. The average time of intensification of symptoms was 3.5 days (2–5 days) after the surgery. During slit lamp examination, 1 × 1 mm paracentral corneal infiltrate was found in two patients (Figure 1a,b), and paracentral infiltrate of about 3 × 2.5 mm in the next two patients was found (Figure 2a,b). Infiltrates were not staining with fluorescein (Figure 3a). Moreover, delayed healing of epithelium in one patient occurred in the form of stained erosion (Figure 3b). Swabs and scrapings were taken from all patients to exclude infectious corneal infiltrate. Both were taken directly onto blood agar and Sabouraud medium with a coarse brush to detect bacterial and fungal infections. The results of microbiological tests were negative. During the first check-up, each patient had the eye shield removed and treatment with levofloxacin and dexamethasone was continued five times a day until smear results were ready. After the results were obtained, patients with smaller infiltrate had the antibiotic reduced to three times a day, and dexamethasone drops increased to seven times a day. In patients with greater infiltration, an increased dose of the antibiotic was continued for a certain period due to the suspicion of infectious infiltrate despite a negative culture result, and an increased dose of the steroid was administered with great caution and required more frequent check-ups at the clinic (every three days). Detailed times of treatments and symptom relief are presented in Table 1. Inflammatory infiltrates resolved in all of the patients observed. In two patients, the withdrawing infiltrate left opacification in the stroma. 

In vivo confocal microscopy shows stimulated epithelial cells with round inflammatory cells and Langerhans cells of varied maturity at the site of ulceration [13]. These cells are located mainly in the deep layers of the epithelium and the Bowman’s membrane (Figure 4 and Figure 5). In stroma, there are stimulated keratocytes producing substances that cause stromal fibrosis, resulting in the formation of a scar (Figure 6). The OCT revealed hyperreflective lesions of various size and depth which disappeared or decreased over time (Figure 7). Best-corrected visual acuity did not deteriorate after the procedure, and parameters of corneal curvature and astigmatism decreased. It is probable to assume that this is related to the fact that the infiltrates were not located centrally (Table 2). The patients did not report comorbidities or allergies. Due to the young age, inadequate hygiene conditions and inappropriate postoperative procedures not in accordance with the recommendations given by the authors of the present article cannot be excluded. The entire procedure was performed in accordance with the CXL treatment guidelines, each patient was operated on by a different doctor, and riboflavin as well as instruments used for the procedure came from different lots. One patient had to undergo CXL on the other non-operated eye due to the significant progression of keratoconus. Due to the fact that sterile infiltrate occurred in the patient’s non-operated eye, the decision was made to take special care and introduce additional protection before the procedure. It consisted of the use of eyelid hygiene for a period of two weeks before the procedure, and dexamethasone drops (1 mg/mL) three times a day for a week before the procedure. The patient underwent the operation with success, and the eye was healed seven days after the procedure with no changes to the cornea (Figure 8). Most likely, such preventive measures can prepare the patient for the procedure, and in predisposed individuals, reduce the immune reaction immediately after the procedure. 

## 4. Discussion

Despite new reports on cases of patients with sterile corneal infiltrates after CXL procedures, their small number makes it difficult to determine the etiology and mechanism of their formation. Over the years, several authors have put forward several hypotheses. According to the literature, 87 people with this complication after the CXL procedure have been described so far [7,8,9,10,11,14,15,16,17,18,19,20,21,22,23]. Comparing this to the number of CXL procedures performed in the world, it is a rare complication. However, it cannot be underestimated, and—according to the authors of this article—quick diagnosis has an impact on the method of treatment and possible late complications. The first case was described in 2009 by Angunawela et al. [7]. He advanced a hypothesis that in patients post-CXL procedure, the phenomenon of increased cellular resistance (CMI) to the Staphylococcus aureus antigen occurs in the area with increased amounts of tears by the eye shield. The author himself followed the example of a study in which a similar response was observed in patients with sterile peripheral keratitis who suffered from blepharitis [24]. Today, this hypothesis is still valid, and many authors use it not only post-CXL but also after refractive procedures [23,25]. Recent reports suggest a strong relationship between vernal keratoconjunctivitis and Down’s syndrome being risk factors for the development of sterile corneal infiltrates [8]. Vigorous rubbing of eyes, inadequate hygiene of the eyelids margin, as well as their impaired function, may lead to a greater accumulation of staphylococcal exotoxin on their margins. In the study described here, patients did not report allergies nor were they diagnosed with vernal keratoconjunctivitis. Nevertheless, frequent rubbing of the eyes and poor hygiene cannot be excluded, especially since the complication concerned young men who were reluctant to describe their ailments. Therefore, in one of the patients who underwent the CXL procedure on the second eye, after previous keratitis in the first eye, the decision was made to take special care and introduce additional anti-inflammatory protection. Therefore, it seems reasonable to carry out all the steps before the CXL procedure so that the eye surface is stable and the immune induction after the procedure as small as possible. Preventive measures were taken by applying eyelid margin hygiene two weeks before the procedure and the application of a steroid in the form of drops seven days before the procedure. 

When analyzing all the reports to date, it is noteworthy that the largest reported groups of patients with sterile corneal infiltrate after CXL had already undergone the accelerated A-CXL procedure. This was first described in 2016 by Çerman: 19 cases, 3 of which underwent the standard procedure and 16 the A-CXL. The authors concluded that this difference was not statistically significant, and indicated epithelial damage and postoperative use of NSAIDs as the main cause of infiltration. Another seven patients were described in 2020 by Kodavoor; the results suggested that further studies of complications following A-CXL treatment, including prospective studies, should be conducted. In the same year, Çakmak described another group of 25 patients after A-CXL procedure. This study suggested that the phototoxicity of the CXL procedure may have an overwhelming effect on the cause of sterile corneal infiltrate. He thus confirmed the observations from 2012 by Ghanem. He concluded that this effect induces an enhanced immune response by recognizing the native proteins of the patient as foreign ones, which is induced directly by exposure to UV. This is due to the change in the antigenic characteristics after the procedure. In 2014, Lam reported that cornea with a thickness of <425 µm and a maximum corneal curvature of >60 D may have an influence on the greater cytotoxic effect of UVA radiation on the cornea. He paid particular attention to the working distance of the UVA lamp during the procedure, which, with slight changes in position, may exceed the toxicity threshold for the corneal endothelium. It should be noted that the irradiation dose may have a toxic effect, therefore each patient should be individually assessed for the procedure by adjusting the radiation dose and choosing an appropriate CXL procedure.

In the clinic, Epi-off and Epi-on procedures are performed. The role of the epithelium appears to be very important in the occurrence of corneal infiltrates. Probably, the mechanism of epithelial damage and its subsequent healing may induce an increased and inadequate immune response. When the corneal epithelium is damaged, many growth factors and cytokines are released. These factors play an essential role in the interaction of epithelium with stroma and the successful healing of the wound [26]. The possible cytotoxic effect of the procedure and the individual predisposition of a given patient in combination with mechanical damage to the epithelium may have an influence on the induction of increased immune response and sterile infiltrate of the cornea. 

Despite the fact that the incidence of this complication is low, every effort should be made to ensure that patients are properly assessed for CXL procedures. Previous preventive measures before the procedure (stabilization of the corneal surface), during the procedure (sterile surgical field, meticulous performance of surgical procedures) and after the procedure (maintaining a high degree of postoperative care in patients from the risk group) may have a significant impact on reducing the incidence of this complication.

## 5. Conclusions

In conclusion, epithelial damage, impaired regeneration and the associated inflammatory response seen in IVCM may be the basis for the formation of sterile infiltrates. Regardless of the final appearance at biomicroscopy and confocal microscopy, the infectious origin of the molded corneal infiltrates and the relative inflammatory response can never be excluded. These occurrences are furthermore linked to the patient’s postoperative compliance and to environmental/hygienic factors, including the therapeutic contact lens. In consideration of its antimicrobial nature, the cross-linking procedure itself is never a direct cause of infection, but it can cause a sterile inflammatory reaction and wound healing activation [13]. This study has some limitations due to its retrospective nature. Prospective studies, especially after the A-CXL procedure, could provide more knowledge about the formation of sterile infiltrates.

## Figures and Tables

**Figure 1 jcm-11-00585-f001:**
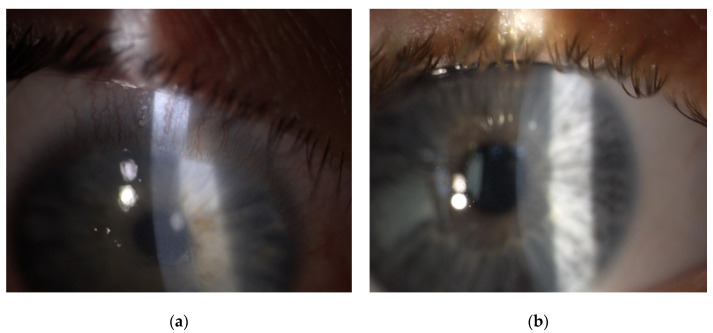
Slit-lamp postoperative images showing sterile keratitis seven days after the CXL procedure due to progressive keratoconus. (**a**) a 16-year-old male patient with paracentral corneal infiltrate and conjunctival hyperemia. (**b**) a 17-year-old male patient with paracentral corneal infiltrate.

**Figure 2 jcm-11-00585-f002:**
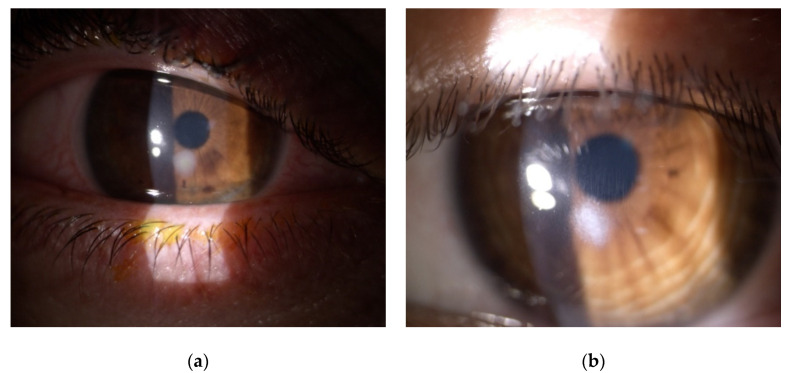
Slit-lamp postoperative images showing sterile keratitis seven days after CXL procedure due to progressive keratoconus. (**a**) a 28-year-old male patient with paracentral corneal infiltrate and significant conjunctival injection. (**b**) a 20-year-old male patient with paracentral corneal infiltrate.

**Figure 3 jcm-11-00585-f003:**
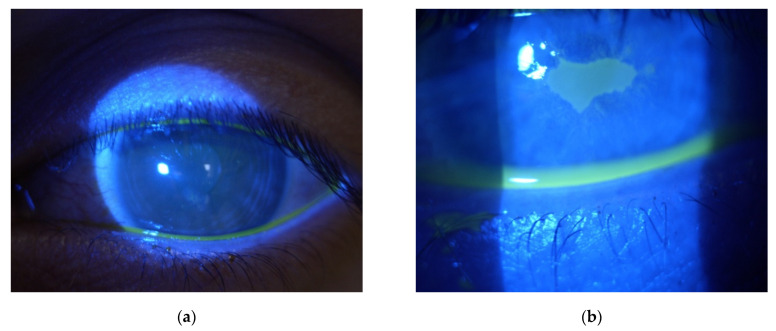
Slit-lamp postoperative images showing corneal fluorescein staining seven days after the CXL procedure due to progressive keratoconus. (**a**) a 20-year-old male patient with paracentral sterile corneal infiltrate not staining with fluorescein (**b**) a 16-year-old male patient with paracentral corneal infiltrate not staining with fluorescein, who additionally experienced corneal epithelialization in the form of erosion of fluorescein-staining cornea.

**Figure 4 jcm-11-00585-f004:**
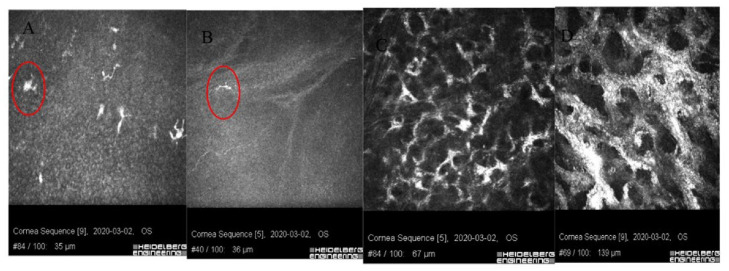
Images of confocal microscopy of patients with sterile corneal infiltrate after cross-linking procedure. (**A**) In deep layers of the epithelium, numerous LG cells of varied maturity (circle) are visible. (**B**) Single inflammatory cells (circle) visible in the layer of Bowman’s membrane with reduced SNP plexus. (**C**) Site of ulceration with stimulated keratocytes forming a characteristic honeycomblike network. (**D**) Hyperreflective tissue surrounding keratocytes corresponding to fibrosis.

**Figure 5 jcm-11-00585-f005:**
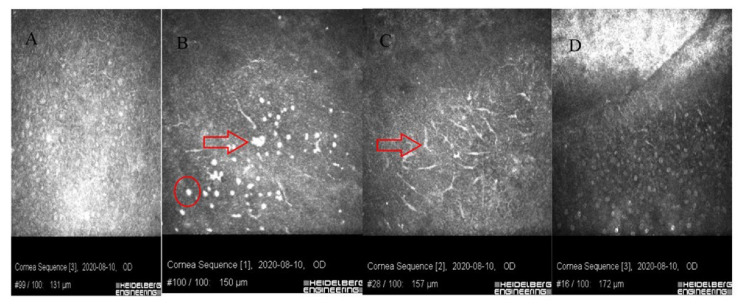
Images of confocal microscopy of patients with sterile corneal infiltrate after cross-linking procedure. (**A**) Stimulated epithelial cells with visible cell nuclei. (**B**) In the layer of epithelial cells, there are round inflammatory cells (circle), single Langerhans cells and hyperreflective apoptotic epithelial cells (larger than the inflammatory cells-arrow). (**C**) In deep layers of the epithelium and Bowman’s layer, there are numerous Langerhans cells with protrusions (arrow). (**D**) Oblique scan-from bottom, a layer of stimulated epithelium through the Bowman’s layer with LG cells, hyperreflective fibrous tissue visible in anterior stroma (from top).

**Figure 6 jcm-11-00585-f006:**
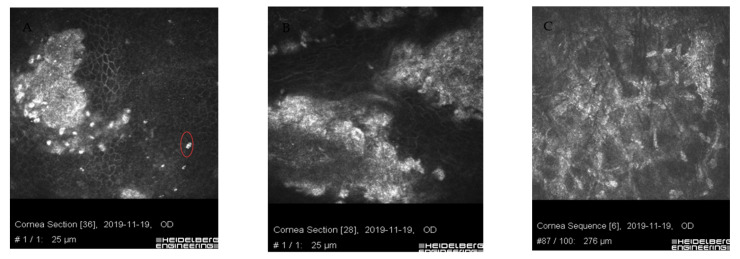
Images of confocal microscopy of patients with sterile corneal infiltrate after cross-linking procedure. Residual form of keratitis (**A**,**B**) with hyperreflective tissue corresponding to scar tissue, irregularly structured epithelial cells with impacted apoptotic hyperreflective cells (circle). (**C**) Corneal stroma layer with hyperreflective structures corresponding to fibrosis and normal keratocyte nuclei.

**Figure 7 jcm-11-00585-f007:**
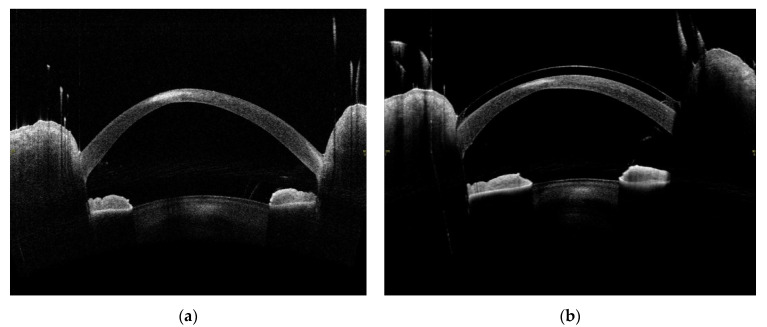
Images created by the use of optical computed tomography of the anterior segment of the eye of a 20-year-old male patient with peripheral corneal infiltrate after the CXL procedure. (**a**) Hyperreflective corneal infiltrate seven days after the procedure, 1.83 mm deep and 1.392 mm wide. (**b**) Hyperreflective residual scar in the corneal stroma six months after the procedure, 0.092 mm deep and 1.323 mm wide.

**Figure 8 jcm-11-00585-f008:**
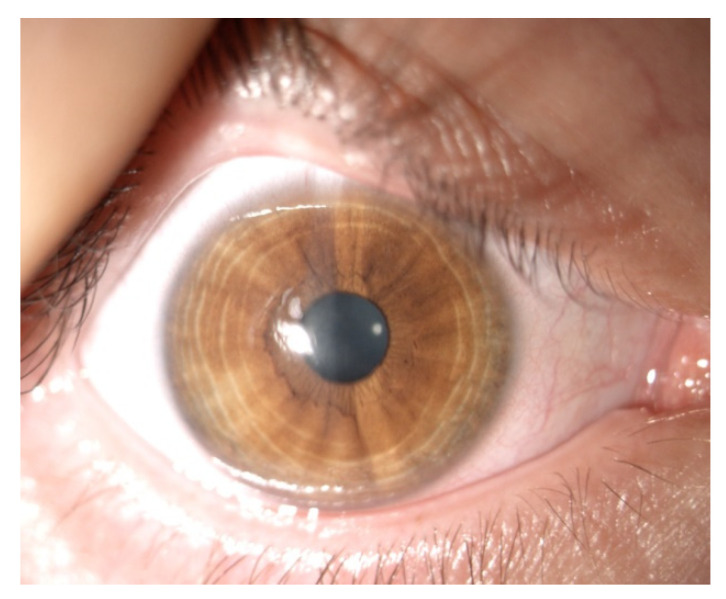
Image from a slit lamp of the right eye of a 20-year-old male patient, seven days after CXL procedure, for whom preoperative preventive measures were applied due to previous sterile corneal inflammation in left eye. Epithelial cornea, no signs of inflammation.

**Table 1 jcm-11-00585-t001:** Detailed description of patients with sterile corneal infiltrates.

Patient	1	2	3	4
Age	16	17	20	28
Gender	m	m	m	m
Microbiology swab test result	negative	negative	negative	negative
Site of infiltrate	paracentral	paracentral	paracentral	paracentral
Size of infiltrate (mm)	1 × 1	1 × 1	3 × 2.5	3 × 2.5
Exacerbation of local symptoms (day)	3	5	4	5
First check-up(day)	6	7	7	7
Time of treatment (weeks)	6	4	12	8
Scar	none	none	present	present
Symptoms of allergy, VKC	none	none	none	none

VKC—Vernal keratoconjunctivitis.

**Table 2 jcm-11-00585-t002:** Preoperative corneal topographic parameters based on Scheimpflug camera analysis.

Patient	1	2	3	4
Degree of keratoconus (Amsler–Krumeich classification)	3	3	3–4	2–3
Preoperative corneal thickness (µm) max (min)	475 (449)	483 (476)	508 (469)	561 (518)
K1 [D]	44.1	50.1	53.2	41.3
K2 [D]	49.0	51.8	58.6	45.5
Kmean [D]	46.4	50.9	55.7	43.3
Astig [D]	4.9	1.7	5.4	4.1
BCVA 6 M-postop	0.3	0.3	0.6	0.4

## Data Availability

The data used to support the findings of this study are included in the article. The data will not be shared due to third-party rights and commercial confidentiality.

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
