# Peer review of "Retrospective Analysis of Sterile Corneal Infiltrates in Patients with Keratoconus after Cross-Linking Procedure"

_jcm, 2022, doi:10.3390/jcm11030585_

Round 1

Reviewer 1 Report

  • Page 1, Line 34, “One of the methods to inhibit progression of the disease is the cross-linking of collagen fibres within the cornea (Corneal cross-linking, CXL), first described by Wollensak et al. [2].” I am unaware of other methods that have consistently been proven to inhibit progression. I would either provide a reference or rephrase.
  • Page 1, Line 41. “exposure to the laser beam”. The UV light source in CXL is not a laser. It does not have the same properties and its purpose is completely different of, say, excimer laser. This is a serious misconception and the authors should correct it.
  • Page 1, Line 45 “lowering their optical properties”. What do the authors mean by this? If they are referring to a specific index that can be measured as a continuous variable they should state which one. If not they should avoid using “lowering”.
  • Page 2, Line 52. “after the accelerated CXL procedure and possibly an increase in exposure 52 to the UV dose (11,13)”. The authors provide 2 references for this sentence. The first one is a 2016 study that directly states that there was not an increase of sterile infiltrates associated with ACXL. The authors also state exactly that in the Discussion section. The second one is a 2020 study that includes only patients who performed ACXL, so no comparison is made. The authors should either correct this sentence or better justify their conclusions.
  • Page 2, Line 92. “The average time from procedure to onset of symptoms was 3.5 days (two to five days).” I’m a bit puzzled by this statement because virtually all my epi-off patients have pain and redness from day 0 after CXL. These are the normal symptoms after CXL. Did the symptoms improve and then suddenly worsened? The authors should clarify what they mean by this.
  • Page 3, Line 94 and 95. The authors use “1x1” and “3x2.5”. I’m assuming they mean millimeters but it should be stated.
  • Page 5, Table 1. The authors use the term “Symptoms of infection” but it is not necessarily true that it is an infection.
  • Page 5, Line 154. “We excluded the iatrogenic factor.” What do the authors mean by this sentence? Sterile corneal infiltrates after CXL is quite obviously an iatrogenic condition. I assume they mean that there was no negligent conduct from the medical staff but those 2 concepts are not synonyms. Please rephrase or explain.
  • Page 9, Line 208. The authors mention that “there have been more and more reports presenting cases of patients with sterile corneal infiltrate after the CXL” while it is stated in the Introduction section that “Few papers describing this complication have been published over the years”. While both sentences are subjective opinions they nonetheless sound contradictory when coming from the same authors. I would rephrase.
  • Page 9, Line 214. “quick diagnosis has an impact on the method of treatment and possible late complications.” The authors should either provide a reference for this claim or state that it is the author’s opinion.
  • Page 10, Line 241. “Another large group of seven patients was described by Kodavoor in 2020, suggesting that a small number of studies on complications after A-CXL procedure and prospective studies should be implemented.” Really confusing english, I would try and improve on this sentence
  • Page 10, Line 246, “He was the first to present this hypothesis by Ghanem in 2012.” I don’t understand this sentence. Was he the first to reaffirm the hypothesis that had already been proposed? The sentence looks nonsensical as it is.
  • Page 10, Line 258. “Not all cases, both in our centre and in others showed incidences of sterile corneal infiltrates after the CXL transepithelial procedure.” Again, poor english construction. Does it mean that there were no cases with transepithelial CXL? I would try and make the sentence easier to understand.
  • Page 10, Line 271. “In conclusion, epithelial damage and photo-toxic effect may have a major influence on the formation of sterile infiltrates”. The first sentence of the Conclusion section should contain the main findings of a study. Even if the effect of epithelial damage and phototoxicity is real, it is hard to argue that there was compelling evidence for it in the results of this study, since no attempt to study that was made. I don’t feel this should be the main conclusion of the study because it simply is not backed by the results shown (regardless if it’s true or not)

Author Response

Dear Reviewer,

Thank you very much for your interest in our manuscript entitled Retrospective Analysis of Sterile Corneal Infiltrates in Patients with Keratoconus after Cross-linking Procedure.

We are grateful for your helpful comments and for recognizing the value of our study. The suggested changes have made our paper more interesting, valuable, and suitable for publication.

Answers to comments:

As suggested by you, several sentences have been rephrased to make the text clearer and easier to understand. Statements that were not clear, especially in the introduction and discussion sections, have been corrected, so that they are more transparent for the reader.

Sequentially presenting:

  • Page 1, Line 34, “One of the methods to inhibit progression of the disease is the cross-linking of collagen fibers within the cornea (Corneal cross-linking, CXL), first described by Wollensak et al. [2].” I am unaware of other methods that have consistently been proven to inhibit progression. I would either provide a reference or rephrase.

The sentence quoted above has been changed to:

“A method of treatment with proven effectiveness is the cross-linking of collagen fibers within the cornea (Corneal cross-linking, CXL), first described by Wollensak et al. [2].”

  • Page 1, Line 41. “exposure to the laser beam”. The UV light source in CXL is not a laser. It does not have the same properties and its purpose is completely different of, say, excimer laser. This is a serious misconception and the authors should correct it.

We acknowledge the error. The adequate fragment in the sentence has been changed to :

“to the UV light source”.

  • Page 1, Line 45 “lowering their optical properties”. What do the authors mean by this? If they are referring to a specific index that can be measured as a continuous variable they should state which one. If not they should avoid using “lowering”.

            The adequate fragment in the sentence was changed to:

            “and loss of its optical properties”

  • Page 2, Line 52. “after the accelerated CXL procedure and possibly an increase in exposure 52 to the UV dose (11,13)”. The authors provide 2 references for this sentence. The first one is a 2016 study that directly states that there was not an increase of sterile infiltrates associated with ACXL. The authors also state exactly that in the Discussion section. The second one is a 2020 study that includes only patients who performed ACXL, so no comparison is made. The authors should either correct this sentence or better justify their conclusions.

We have decided to remove this sentence from the introduction. The probable factor, which is an increase of sterile infiltrates associated with A-CXL, is presented in the discussion part of the article and is based on the analysis of existing scientific reports and the number of cases of sterile infiltrates after the conventional CXL procedure compared with the accelerated A-CXL procedure. It is not supported by any evidence, just our personal reflection on the matter. Indeed, in the first reference the authors write that the results are not statistically significant, but the presented results, in which the total number of sterile infiltrates of 19 cases, out of which 3 are after the standard procedure and 16 after the accelerated procedure, give us food for thought.

  • Page 2, Line 92. “The average time from procedure to onset of symptoms was 3.5 days (two to five days).” I’m a bit puzzled by this statement because virtually all my epi-off patients have pain and redness from day 0 after CXL. These are the normal symptoms after CXL. Did the symptoms improve and then suddenly worsened? The authors should clarify what they mean by this.

This is true. In our clinic, immediately after the CXL procedure, patients also report pain, redness of the eye, and decreased vision. These symptoms should gradually disappear over time; however, in patients with sterile corneal infiltrates this was not the case.

The sentence indicated by you has been changed to:

“The first check-up was scheduled 7 days after the surgery. Only 1 patient came in a day earlier than the  designated appointment; the rest of the patients attended on the appointed date reporting unrelenting redness, pain in the eye that had undergone the procedure, and progressive deterioration of vision. The average time of intensification of symptoms was 3.5 days (2-5 days) after the surgery.”

  • Page 3, Line 94 and 95. The authors use “1x1” and “3x2.5”. I’m assuming they mean millimeters but it should be stated.

Yes it is about millimeters, we have made the appropriate corrections.

  • Page 5, Table 1. The authors use the term “Symptoms of infection” but it is not necessarily true that it is an infection.

The phrase has been changed to:

            “Exacerbation of local symptoms”

  • Page 5, Line 154. “We excluded the iatrogenic factor.” What do the authors mean by this sentence? Sterile corneal infiltrates after CXL is quite obviously an iatrogenic condition. I assume they mean that there was no negligent conduct from the medical staff but those 2 concepts are not synonyms. Please rephrase or explain.

The sentence has been changed to changed to:

“The entire procedure was performed in accordance with the CXL treatment guidelines, each patient was operated on by a different doctor, and riboflavin as well as instruments used for the procedure came from different lots”.

  • Page 9, Line 208. The authors mention that “there have been more and more reports presenting cases of patients with sterile corneal infiltrate after the CXL” while it is stated in the Introduction section that “Few papers describing this complication have been published over the years”. While both sentences are subjective opinions they nonetheless sound contradictory when coming from the same authors. I would rephrase.

The sentence has been changed to:

”Despite new reports on cases of patients with sterile corneal infiltrates after CXL procedures, their small number makes it difficult to determine the etiology and mechanism of their formation”.

  • Page 9, Line 214. “quick diagnosis has an impact on the method of treatment and possible late complications.” The authors should either provide a reference for this claim or state that it is the author’s opinion.

The sentence has been changed to:

“However, it cannot be underestimated, and – according to the authors of this article – quick diagnosis has an impact on the method of treatment and possible late complications”.

  • Page 10, Line 241. “Another large group of seven patients was described by Kodavoor in 2020, suggesting that a small number of studies on complications after A-CXL procedure and prospective studies should be implemented.” Really confusing english, I would try and improve on this sentence

The sentence has been changed to:

“Other seven patients were described by Kodavoor in 2020; the results suggested that further studies of complications following A-CXL treatment, including prospective studies, should be conducted”.

  • Page 10, Line 246, “He was the first to present this hypothesis by Ghanem in 2012.” I don’t understand this sentence. Was he the first to reaffirm the hypothesis that had already been proposed? The sentence looks nonsensical as it is.

The sentence has been changed to:

“He thus confirmed the observations by Ghanem from 2012”.

  • Page 10, Line 258. “Not all cases, both in our centre and in others showed incidences of sterile corneal infiltrates after the CXL transepithelial procedure.” Again, poor english construction. Does it mean that there were no cases with transepithelial CXL? I would try and make the sentence easier to understand.

The sentence has been changed to:

“The role of the epithelium appears to be very important in the occurrence of corneal infiltrates”.

  • Page 10, Line 271. “In conclusion, epithelial damage and photo-toxic effect may have a major influence on the formation of sterile infiltrates”. The first sentence of the Conclusion section should contain the main findings of a study. Even if the effect of epithelial damage and phototoxicity is real, it is hard to argue that there was compelling evidence for it in the results of this study, since no attempt to study that was made. I don’t feel this should be the main conclusion of the study because it simply is not backed by the results shown (regardless if it’s true or not)

The sentence has been changed to:

“In conclusion, epithelial damage, impaired regeneration, and the associated inflammatory response seen in IVCM may be the basis for the formation of sterile infiltrates”.

The manuscript has been verified for correctness of the English language. The sentences transformed stylistically to avoid the use of “we” or “our” have been marked with green font. The other ones, marked in red, indicate the sentences transformed to meet the requirements of both Reviewers.

Again, we would like to thank the Reviewer for helpful comments, and for recognizing the value of our paper.

Reviewer 2 Report

This is an interesting study that concerns rare cases of adverse events following cross linking. It is important to document these kinds of  events and this article contributes to that. 
That said, I would not call this study retrospective, since no indication of the method of case selection was given. There was no indication that patients gave informed consent for the review of their records.  No ehtics committee approval. It would be better to focus on the 4 cases and present it as a Case Series. Especially since, out of the 500 or so charts looked at, it may be possible that small infiltrates were missed or not noted, as the methods of examination and analysis are not similar from one practitioner to another.  Presenting in a case series format is better and does not make the article less relevant. 
Some remarks: 
- Do not use the form of "we" "our" - use a more generic form in the text. 
- Justify the fact that the first check-up is at 7 days which seems long compared to the usual standards 
- Justify the fact that the patients did not consult at the time of the onset of symptoms. An eye that hurts after an operation should be seen immediately. Among other things, this could have made a difference for patients with larger infiltrates
- Describe swabbing and scrapping methods. No details are provided. Also identify methods of culture (no details provided). 
- Biases should not be in the conclusion but in the last paragraph of the text. 
- Indicate if patients were contact lens wearers prior to cross linking. This can have a significant impact 
- Justify the use of Air optix as a dressing lens. It is not recognized by the FDA as a dressing lens. 
-The material of Air Optix is known to be more associated with GPC or other inflammatory reactions. This may be a factor to consider. 
- Indicate the method of application of the bandage lens to the patient's eye. Bacterial exotoxins may occur from contamination of the lens during application. Especially if the lens is picked up from the case with a finger and not sterile forceps. Because the finger touches the back of the lens and the lens comes into contact with a debrided cornea, infiltrates can more easily develop. 
- Indicate the product used to lubricate the eye during the days the lens was worn 
- Indicate if written instructions regarding eye hygiene and care were given after surgery. 

With these details, the article should be more informative. 

Author Response

Dear Reviewer,

Thank you very much for your interest in our manuscript entitled Retrospective Analysis of Sterile Corneal Infiltrates in Patients with Keratoconus after Cross-linking Procedure.

We would like to thank you for your helpful comments, and for recognizing the value of our study. We believe that the suggested changes as well as the respective corrections introduced to the text have made our paper more interesting, valuable, and suitable for publication.

Answers to comments:

The study is part of many-year monitoring of patients after CXL; therefore, the term retrospective appears.

We did obtain approval from the bioethics committee. The approval is included at the end of the article in the following wording: “Institutional Review Board Statement:  The study was conducted according to the guidelines of the Declaration of Helsinki, and approved by the Bioethical Committee of the Medical University of Silesia, Katowice, Poland (PCN/0022/KB1/21/21)”. Additionally, we have decided to include this information in Materials and Methods.

In our clinic, the procedure of qualification for surgery as well as the first check-up after the CXL procedure has been performed every time by the same doctor (Ewa Wróblewska-Czajka, MD) since 2011 until today. Moreover, the fact that over a period of 10 years 4 sterile infiltrations occurred while the CXL procedure was performed 543 times confirms that this is a very rare complication.

For each surgical procedure, patients receive an informed consent form with a detailed description of the procedure as well as information about the treatment after it. Despite this, only 1 patient attended the follow-up appointment one day earlier than the indicated timing suggested. The rest of the patients on day 7 after the procedure, despite the occurrence of worrisome symptoms!

Information on the method of collecting the material for microbiological tests has been added as per your suggestion.

Indeed, majority of patients wore contact lenses before the procedure, and were advised to discontinue wearing them 14 days prior to surgery.

After the procedure, the contact lenses were applied sterile in the operating room.

Thank you for your valuable comment on the Air Optix lens, we admit that its impact has not been analyzed. We will change CL for future procedures.

Some sentences have been rephrased according to your comments. The ones transformed stylistically to avoid the use of “we” or “our” have been marked with green font. The other ones, marked in red, indicate the sentences transformed to meet the requirements of both Reviewers.

The manuscript has been verified for correctness of the English language.

Again, we would like to thank you for your helpful comments, and for recognizing the value of our paper.

Round 2

Reviewer 2 Report

THanks to the authors for the modifications made to the manuscript. It is a better article now. 

Author Response

This manuscript is a resubmission of an earlier submission. The following is a list of the peer review reports and author responses from that submission.